# Unraveling Epigenetic Changes in *A. thaliana* Calli: Impact of HDAC Inhibitors

**DOI:** 10.3390/plants12244177

**Published:** 2023-12-15

**Authors:** Pavlína Pírek, Karolína Kryštofová, Ingrid Kováčová, Anna Kromerová, Dagmar Zachová, Ondřej Helia, Klára Panzarová, Jiří Fajkus, Zbyněk Zdráhal, Gabriela Lochmanová, Miloslava Fojtová

**Affiliations:** 1Mendel Centre for Plant Genomics and Proteomics, Central European Institute of Technology, Masaryk University, 62500 Brno, Czech Republic; pavlina.pirek@ceitec.muni.cz (P.P.); karolina.krystofova@ceitec.muni.cz (K.K.); ingrid.kovacova@ceitec.muni.cz (I.K.); dagmar.zachova@ceitec.muni.cz (D.Z.); fajkus@sci.muni.cz (J.F.); zdrahal@sci.muni.cz (Z.Z.); 2National Centre for Biomolecular Research, Faculty of Science, Masaryk University, 62500 Brno, Czech Republic; 499115@mail.muni.cz; 3PSI (Photon Systems Instruments), spol. s.r.o., 66424 Drásov, Czech Republic; panzarova@psi.cz

**Keywords:** *Arabidopsis thaliana*, callus formation, epigenetics, histone posttranslational modifications, mass spectrometry, trichostatin A, sodium butyrate

## Abstract

The ability for plant regeneration from dedifferentiated cells opens up the possibility for molecular bioengineering to produce crops with desirable traits. Developmental and environmental signals that control cell totipotency are regulated by gene expression via dynamic chromatin remodeling. Using a mass spectrometry-based approach, we investigated epigenetic changes to the histone proteins during callus formation from roots and shoots of *Arabidopsis thaliana* seedlings. Increased levels of the histone H3.3 variant were found to be the major and most prominent feature of 20-day calli, associated with chromatin relaxation. The methylation status in root- and shoot-derived calli reached the same level during long-term propagation, whereas differences in acetylation levels provided a long-lasting imprint of root and shoot origin. On the other hand, epigenetic signs of origin completely disappeared during 20 days of calli propagation in the presence of histone deacetylase inhibitors (HDACi), sodium butyrate, and trichostatin A. Each HDACi affected the state of post-translational histone modifications in a specific manner; NaB-treated calli were epigenetically more similar to root-derived calli, and TSA-treated calli resembled shoot-derived calli.

## 1. Introduction

The exceptional capacity of plants to regenerate after damage has numerous applications in plant biotechnologies and breeding programs, including targeted genome modifications. Importantly and in contrast to animals, this process may lead not only to the repair of the damage or wounding but also to the regeneration of the new plant (reviewed in [1]). For all types of higher plant regeneration, i.e., tissue regeneration, de novo organogenesis, and somatic embryogenesis, marked progress has been achieved in the characterization of complex changes associated with differentiation/dedifferentiation processes. The formation of pluripotent callus from plant tissue or organs is the first step of de novo organogenesis. Callus induction and subsequent regeneration of shoots are based on specific ratios of plant hormones (auxins and cytokinins) in cultivation media [2]. Calli are induced from plant organs/tissues using a callus-inducing medium (CIM) with high auxin and low cytokinin concentrations, and shoots are regenerated following a change in the phytohormone ratio in favor of cytokinins. Roots regenerate on the medium with auxin and low or no cytokinins [3]. Despite the textbook definition of the callus as a mass of undifferentiated totipotent cells, callus formation is more likely likened to the formation of a lateral root. This hypothesis is supported by the results of physiological studies demonstrating that xylem-pole pericycle cells are critical for both callus and lateral root initiation [4,5], as well as genome-wide transcriptomic analyses showing the similarity between callus and root apical meristem transcriptional profiles, even those of calli derived from aerial plant organs [6]. In this respect, the callus is considered a mass of pluripotent cells.

Extensive changes in gene expression—over 10,000 differentially expressed genes were identified in leaf explant-derived callus [7]—are correlated with the dynamic modulation of the chromatin structure, including changes in epigenetic patterns. Epigenetic modifications of chromatin components, i.e., methylation of DNA and various marks decorating amino-terminal parts of histones, play an important role in determining the chromatin structure and thus influence the transcription activity of genes located in the respective region (reviewed in [8,9]). Within histones, specific amino acids, mainly lysine, arginine, and serine, are post-translationally modified by methylation, acetylation, phosphorylation, and other types of functional groups. Combinatorial patterns of epigenetic chromatin marks generate “codes” for the recruitment of protein complexes that modulate the chromatin structure and play roles in DNA transcription and repair [10]. During callus culture establishment, global chromatin decondensation coupled with local chromatin compaction was observed [11]. Modulation of the activity of enzymes responsible for loading and erasing chromatin epigenetic marks, histone acetyl/methyltransferases, and histone deacetylases/demethylases using mutant plants or application of specific inhibitors demonstrated the importance of epigenetic modifications for cellular reprogramming during callus formation. A distinct example is H3K27me3, a repressive modification that decorates developmentally silenced genes. *Arabidopsis thaliana* mutants with loss of function of respective histone methyltransferase(s) lost cell identity and spontaneously formed callus-like or somatic embryo-like structures [12,13]. Specific H3K27me3 changes were observed during the early phases of callus cultivation: H3K27me3 loading on leaf identity genes and the opposite process on the genes determining root identity [14].

Histone acetylation is correlated with the formation of an open chromatin structure and, thus, transcriptional activity. Historically, this was attributed to the elimination of the positive charge of lysine amino acid residue, which is the preferential target for acetylation, and thus weakening histone–DNA interactions. Recent hypotheses prefer an explanation that also considers the binding and subsequent activity of specific protein readers as it is not probable that relatively subtle changes in the charge of N-terminal parts of histones influence the chromatin structure (reviewed in [15]). Considering the shift to structurally more accessible chromatin during callus formation, this process is correlated with the gain of acetylated histone forms. Nevertheless, the upregulation of histone deacetylase-coding genes [16], as well as activation of histone acetyltransferases [17], was shown to be essential for epigenetic reprogramming during the establishment of callus culture, evidencing that multiple and probably functionally diverse signaling pathways are regulated by this histone modification during cellular dedifferentiation.

As an alternative to the analysis of mutant plants with loss of function of genes encoding proteins involved in epigenetic processes, exposure to chemicals acting as specific modulators of the activity of these enzymes is used in functional studies. With respect to acetylation, trichostatin A (TSA), and sodium butyrate (NaB), inhibitors of the RPD3 (reduced potassium deficiency 3) family of histone deacetylases (HDAC) are commonly used, even for clinical purposes (reviewed in [18]). Both these drugs can activate silenced rRNA genes in Arabidopsis [19,20] and induce pleiotropic growth effects during seed germination and seedling development [21,22].

For the analysis of histone post-translational modifications (PTMs), different methodological approaches were used. Among them, mass spectrometry (MS) has been recognized as an important tool. MS enables both qualitative and quantitative untargeted analysis of the levels of histone peptides decorated by the respective set of epigenetic marks. Until recently, these MS-based epigenetic studies were broadly limited to mammalian cells and tissues because of the specificities of plant cells, mainly high levels of secondary metabolites and other compounds contaminating histone extract and lowering the sensitivity and specificity of the analysis. Straightforward MS measurement of histone modifications in plant samples is now possible because of the inclusion of filter-aided sample preparation and histone propionylation in the protocol [23]. Using this approach, modifications of histones were analyzed in *A. thaliana* HDAC mutants, seedlings germinated in the presence of HDAC inhibitors (HDACi) TSA or NaB, and plants grown from these seedlings in soil [24]. As expected, the levels of acetylated histones were significantly higher in the leaves of mutant plants. Notably, these mutants did not exhibit any morphological changes compared with wild-type plants. Surprisingly, changes in acetylation levels were markedly less pronounced in seedlings exposed to HDACi, which exhibited severe phenotypic abnormalities, and fully recovered in the leaves of plants grown from these seedlings in soil. Thus, the strong effects of TSA and NaB on early plant development are not related exclusively to their potency to inhibit RPD3-like histone deacetylases and are of a more general nature [24].

In this study, we induced calli from the shoot and root parts of 7-day-old *A. thaliana* seedlings using standard CIM and medium supplemented with TSA or NaB. We monitored callus formation and PTMs of histones in seedlings and shoot- and root-derived calli. Our results show specific epigenetic patterns in early calli derived from the shoot and root parts of seedlings and their homogenization during long-term callus propagation. Specific contributions of inhibitors of HDAC to the modulation and unification of the early callus histone marks are further manifested.

## 2. Results

### 2.1. TSA Markedly Supported the Callus Formation from Both Shoot and Root Parts of A. thaliana Seedlings

Calli were established from 7-day-old *A. thaliana* seedlings (7ds) using CIM and CIM supplemented with HDACi, 0.5 mM NaB, and 0.5 μM TSA. The seedlings were stretched on plates with the CIM medium carefully to enable phenotype monitoring, and collection of calli derived from the root and shoot parts of the seedlings separately. After 10 days of growth on CIM, microcalli were visible mainly on the upper parts of roots, while the formation of calli from the root tips was less evident (Figure 1A). In calli cultivated for 20 days, root tip-derived calli were clearly visible and callus formation proceeded from the upper and lower parts towards the middle part of the root. NaB exposure induced extensive formation of shoot-derived calli. Root tip formation showed certain heterogeneity with a group of seedlings showing distinct calli induction but the others showed no calli formation even after 20 days of growth (Figure 1A). Importantly, the process of the callus formation was significantly stimulated on both the shoot and root parts of seedlings on the CIM supplemented with TSA (Figure 1A,B). Thus, different extents and patterns of stimulation of the callus formation by HDACi, NaB and TSA, were observed.

Calli derived from root and shoot parts of seedlings were further propagated on CIM for one year. The phenotype of calli was uniform and independent of the callus origin (Figure 1C).

### 2.2. Calli Possess a Higher Level of H3.3 Histone Variant and Distinct Levels of PTMs Compared to Seedlings

Differences in the dynamics of calli formation on CIM and media supplemented with HDACi prompted us to investigate the epigenetic landscape of the calli. For this purpose, we separated the shoot and root part of the seedlings and induced calli from each part separately. Our original goal was to compare the histone status of calli derived from roots (cR) and shoots (cS) with separately analyzed root and shoot parts of 7ds. However, despite several rounds of increasing the amount of roots for histone extraction, we were unable to obtain histone extracts with sufficient protein concentrations. Accordingly, in MS analyses, many post-translationally modified peptide forms were missing in the root samples. Thus, we compared the patterns of histone modifications in calli to merged MS data of root and shoot samples corresponding to whole 7ds. To decipher the reason for the low histone yield from roots, we further compared the data of merged cR and cS (CTR) with 7ds, cS, and cR (Appendix A, sheet “CTR merged data”). Despite enough starting material, a lower overall abundance of cR peptides was detected and the differences between 7ds vs. CTR were found to be more similar to 7ds vs. cS than 7ds vs. cR. Thus, it seems that the low histone yield in roots is related to the nature of the analyzed tissue rather than to the small amount of the input material.

#### 2.2.1. The Levels of H3 Histone Variants in Seedlings and Calli Cultivated for 20 Days

Using an MS-based approach, we characterized the levels of histone H3 variants, H3.1 and H3.3, using their unique peptides K27–R40, i.e., H3.1K27SAPATGGVKKPHR40 and H3.3K27SAPTTGGVKKPHR40. Distinct levels of H3 variants were found between 7ds and calli derived from roots (cR) and shoots (cS). In particular, the H3.1:H3.3 ratio calculated from the median values of replicates was 3:1 in 7ds, while a significant increase in the levels of the H3.3 variant was detected in both cR (H3.1:H3.3 = 1.3:1) and cS (H3.1:H3.3 = 1.8:1; Figure 2A). The impact of NaB treatment on cR (NaB-cR) and cS (NaB-cS) was rather minor and not significant, although the H3.1:H3.3 ratios differed in NaB-cR (H3.1:H3.3 = 2:1) compared to cR. TSA treatment promoted an even higher increase in H3.3 levels, especially in cR (TSA-cR; H3.1:H3.3 = 1.0:1.0) and to a lesser extent in cS (TSA-cS; H3.1:H3.3 = 1.4:1).

#### 2.2.2. The Levels of H3K27 and H3K36 Marks in Seedlings and Calli Cultivated for 20 Days

Overall pattern of K27 and K36 marks on H3 histones

The modulation of H3.1 and H3.3 composition in nucleosomes has a direct impact on PTM status [25,26]. Thus, we quantified histone marks in the H3K27–R40 peptide and compared the methylation and acetylation status between 7ds and calli propagated for 20 days (Figure 2B). Here, post-translationally modified K27 and K36 sites were quantified regardless of their H3.1 or H3.3 origin while the ratio of H3 variants in each sample was considered. The relative abundance of H3K27 and H3K36 marks and respective *p*-values are presented in Appendix A (sheet “2.2.2. H3K27–R40 marks”). Higher levels of H3K36me1 (*p* = 0.015) and H3K36me2 (*p* = 0.069) were found in cR compared to 7ds, while the level of the transcriptionally repressive H3K27me3 mark was lower (*p* = 0.067). A significantly higher level of H3K36me2 (*p* = 0.001) and higher level of H3K27me2 (*p* = 0.087) were detected in cS compared to 7ds. Importantly, we found striking differences in methylation status between cR and cS. In particular, a lower level of K27me2 (*p* = 0.053) together with a higher level of K36me1 (*p* = 0.019) were detected in cR. Such differences in methylation status reflected the distinct levels of the H3.3 variant in cR and cS. Surprisingly, the levels of those histone marks were homogenized in the presence of NaB. The overall PTM patterns of NaB-cR and NaB-cS were comparable (Figure 2B) regardless of the distinct H3.1:H3.3 ratios (Figure 2A).

TSA had a more substantial impact on the PTM status of calli than NaB (Figure 2B). In TSA-cS, increased levels of low-abundant H3K27ac (*p* = 0.045) and H3K36ac (*p* = 0.023) were found, while the overall methylation status resembled that of control cS. On the contrary, the acetylation status in TSA-cR remained at the level of cR but the methylation status was significantly affected, showing lower levels of K27me1 (*p* = 0.015) and K36me1 (*p* = 0.018) together with increased K27me2 (*p* = 0.013) compared to cR. No significant differences in histone mark levels were found between TSA-cS and TSA-cR, indicating that, similarly as in the case of NaB, the homogenization of the PTM profile of calli exposed to TSA occurred regardless of the distinct H3.1:H3.3 ratios. The phenomenon of HDACi-induced PTMs homogenization in root- and shoot-derived calli is evident from the hierarchical clustering of H3K27–R40-modified forms in individual samples (Figure 2B).

Distinct pattern of K27 and K36 marks on H3.1 and H3.3 variants

Next, we determined the levels of PTMs at the K27 and K36 sites for each H3 variant separately. The relative abundance of K27 and K36 histone marks in H3.1 and H3.3 and the respective *p*-values are presented in Appendix A (sheet “2.2.2. H31 and H33 K27-R40 marks”). The data revealed that the difference in PTMs status between cR and cS is mostly driven by distinct K27 and K36 methylation in H3.1 but not in H3.3 (Figure 2C). In particular, higher levels of H3.1K27me1 and H3.1K36me1 (*p* = 0.068 and 0.044, respectively) were observed in cR compared to cS, while no significant differences in the levels of those marks were found in the H3.3 variants. Minor changes in the PTMs status of H3.1K27–R40 were induced by NaB. TSA-cR had significantly lower levels of H3.1K27me1 and H3.1K36me1 (*p* = 0.002 and 0.022, respectively) together with a higher level of H3.1K27me2 (*p* = 0.031) compared to cR. A similar effect of TSA was observed also in H3.3 where, in addition, higher acetylation at K27 was found (*p* = 0.042). In TSA-cS, higher K36ac (*p* = 0.036) was found as the only difference compared to cS (Figure 2C). Hierarchical clustering of H3.1 and H3.3 PTMs levels in individual samples further clearly demonstrated not only differences between cS and cR but also distinct impacts of NaB and TSA on calli formation. Calli, regardless of their origin (root or shoot part of 7ds), grown in the presence of NaB epigenetically resembled cR, while those grown in the presence of TSA were more comparable to cS (Figure 2C).

#### 2.2.3. The Levels of H3 Histone Variants and K27 and K36 Marks in Calli Cultivated for One Year

As calli propagated in CIM medium survived long-term cultivation, we further investigated the dynamics of the epigenetic landscape during long-term passages. Thus, using the same approach, we analyzed 1y-cR and 1y-cS. The H3.1:H3.3 ratios remained unchanged compared to cR and cS (H3.1:H3.3 = 1.4:1 and 1.8:1 in 1y-cR and 1y-cS, respectively; Figure 2D). Importantly, overall changes in H3K27–R40 histone marks led to the homogenization of PTM profiles between 1y-cR and 1y-cS (Figure 2E,F). The relative abundance of H3K27 and H3K36 marks and respective *p*-values are presented in Appendix A (sheet “2.2.3._H3K27-R40 marks”).

### 2.3. Homogenization of the PTM Profiles between Root and Shoot Calli Is a General Effect of HDACi

Given the homogenization of PTM profiles in H3K27–R40 between cR and cS after HDACis treatment, the levels of other H3 and H4 histone marks were investigated to clarify if this finding represents a general pattern or a specific attribute of H3K27–R40. In particular, we evaluated histone marks in H3K9STGGKAPR17 (H3K9–R17), H3K18QLATKAAR26 (H3K18–R26), and H4G4KGGKGLGKGGAKR17 (H4G4–R17) peptides (Figure 3A).

#### 2.3.1. The Levels of H3 and H4 Histone Marks in Calli Cultivated for 20 Days

Distinct modification patterns of H3K9–R17 and H3K18–R26 were found in 7ds compared to cR or cS. The relative abundance of H3 and H4 histone marks and respective *p*-values are presented in Appendix A (sheet “2.3.1._H3-H4_marks”). Generally, lower levels of acetylation (at K9ac, K14ac, K18ac, and K23ac) and K9me2 accompanied by a higher level of K9me1 were found in cR and cS compared to 7ds.

The levels of K9ac (*p* = 0.086) and K9me2 (*p* = 0.080) were higher in cR compared to cS, but the PTMs of H3K18–R26 and H4G4–R17 did not differ between cR and cS. These differences homogenized when the calli were grown in the presence of NaB or TSA, although a slightly higher level of K9ac was still detected in TSA-cS compared to TSA-cR. When we compared the impact of NaB and TSA on histone marks, we found lower levels of K9ac and K9me2 in TSA-treated calli. The most striking difference was detected in the H4G4–R17 peptide where levels of acetylated sites K5ac, K8ac, and K12ac were significantly higher (*p* < 0.05) in the presence of TSA compared to the NaB treatment, whereby the highest acetylation status was observed in TSA-cR (*p* < 0.01). Importantly, such state was related to hyperacetylation in terms of increased levels of tri- and tetra-acetylated H4G4–R17 forms in cR (Appendix A, sheet “2.3.1._H3–H4_forms”).

Hierarchical clustering of H3K9–R17 PTM levels in individual samples showed a similar pattern as in the case of the H3K27–R40 peptide, i.e., differences between cS and cR and epigenetic similarity of NaB- samples with cR and TSA samples with cS (Figure 3A). The differences in the PTM profile of H4G4–R17 between cR and cS were minor but calli exposed to TSA and NaB formed separate clusters that again showed a distinct impact of these drugs on calli formation. The acetylation status of H3K18–R26 was comparable between samples, as evident from the hierarchical relationship (Figure 3A).

#### 2.3.2. The Levels of H3 and H4 Histone Marks in Calli Cultivated for One Year

Surprisingly, when compared 1y-cR and 1y-cS, significantly higher levels (*p* < 0.05) of acetylation marks in histone H3 (H3K14, H3K18, and H3K23 sites) and H4 (H4K5, H4K8, H4K12, and H4K16 sites) were detected in cS (Figure 3B); specifically, hyperacetylation manifested as increased levels of di-acetylated H3K18–R26 and di-, tri-, and tetra-acetylated H4G4–R17 forms in cS. The relative abundance of H3 and H4 histone marks and respective *p*-values are presented in Appendix A (sheet “2.3.2._H3-H4_marks”).

## 3. Discussion

Formation of the pluripotent cell mass, callus, from somatic plant cells is a complex process accompanied by extensive genetic and epigenetic changes [27,28,29]. Generally, active epigenetic marks are enriched and levels of modifications correlated with heterochromatin formation are reduced in callus compared to differentiated plant tissues [30,31]. Based on the transcriptomic data, the initial steps of callus formation share common features with the formation of lateral roots; thus, analysis of calli derived from the shoot (aerial) and root parts of the plant is of high importance. In the study analyzing dynamic transcriptome changes during the initiation of calli from *A. thaliana* root and shoot explants, hundreds of differentially expressed genes were identified within a 96 h time interval [32]. Importantly, the transcription of genes encoding specific epigenetic modifiers, including histone deacetylases, acetyltransferases, and methyltransferases, was down- or up-regulated depending on the phase of the callus induction process. This implies that epigenetic processes are active during callus induction and that the expression of respective enzymes involved in epigenetic regulation is strictly regulated. Distinct epigenetic changes were also detected in analyses of older calli 20 days after their origination (Figure 4). Notably, we found that the ratio of H3.1:H3.3 histone variants decreased significantly during the formation of both types of calli and was maintained at approximately the same level in long-term cultivated calli (Figure 2A,D). H3.1 is a canonical H3 variant and its expression is coupled with DNA replication and replication-dependent chromatin assembly; H3.3 is expressed throughout the cell cycle, is enriched close to the transcription end sites, and is positively correlated with transcription [33,34]. Thus, H3.3 enrichment in calli correlates with the assumption that callus formation is accompanied by both activation and silencing of gene expression, with the former change being predominant. Indeed, the expression of genes necessary for the process of cell dedifferentiation has been linked to the role of a histone regulator of the cell cycle A (HIRA), a chaperone complex that is essential for replication-independent H3.3 loading into chromatin [34].

Next to a significantly higher level of H3.3, a more open chromatin structure in calli compared to differentiated plant tissue is also expressed by a lower level of H3K9me2 repressive mark and a higher level of H3K9me1 (Figure 4A). Reportedly, an increased level of K9me2 was found to associate with enriched H3.1 in heterochromatin regions [35]. Thus, the levels of H3K9 marks found in calli followed the lower level of histone H3.1 but can be further balanced by methyltransferases/demethylases too. Here, we are dealing with the limitations of bottom-up proteomics; we cannot distinguish K9–R17 and K18–R26 of individual H3 variants as those peptides have the same amino acid sequence in H3.1 and H3.3. The only forms with distinguishable origin are K27–R40 peptides. In this case, we can see that each H3 variant is decorated by its own PTM profile, which is further specific for de- and differentiated plant tissues. For instance, ~27% and ~13% of H3.1K27me1 and H3.3K27me1 were detected in 7ds, respectively, which (when considering the H3.1:H3.3 ratio) corresponds to ~23% of the overall K27me1 abundance. Despite the relative decrease in H3.1 in cR, the overall K27me1 level is similar to 7ds due to the increased level of H3.3K27me1 to ~24%. On the other hand, less abundant overall K27me1 found in cS (~18%) is related to changes in K27me1 of both H3 variants. Similarly, as detected in K27–R40 peptides, we suppose that there are distinct PTM states in K9–R17 and K18–R26 of H3.1 and H3.3 variants. The fact that the H3.1:H3.3 ratio substantially contributes to differences in PTM profiles between 7ds and calli propagated for 20 days can explain the observed trend to the lower level of acetylated H3 histone marks (e.g., K9ac, K14ac, K18ac, and K23ac). A statistically significant increase in histone acetylation in young calli samples was not detected in histone H4 either. Our data thus suggest that elevated H3.3 levels are a major factor that can cause chromatin relaxation that does not require additional support through markedly increased acetylation. A different situation was observed in 1-year calli—mainly in 1y-cS—where levels of acetylated histones increased, even above the level detected in seedlings (Figure 3, Appendix A). Notably and similarly to 20-day calli, the levels of acetylated histones were generally higher in 1y-cS than in 1y-cR (Appendix A).

Changes in H3.1K27me1 and H3.3K27me1 were accompanied by a drop in K27me3 levels in cR. For H3K27me3, an epigenetic modification typical for silenced genes, dual effects related to callus formation were reported. Except for the previously mentioned spontaneously formed callus-like and embryo-like structures in H3K27 methyltransferase mutants [12,13], the activity of these enzymes is necessary for the inactivation of leaf identity genes [14]. In our experiments, the level of H3K27me3 was relatively low in all samples (Figure 2B,C; Appendix A). In contrary to cR, H3K27me3 in cS did not drop, which may be correlated with the previously mentioned need to silence leaf identity genes in root-derived callus. Comparison of the methylation levels of lysines located at different histones at different positions showed marked differences between cR and cS, while these were fully comparable in 1-year calli (Figure 2 and Figure 3). This clearly demonstrated the homogenization of the epigenetic pattern in shoot- and root-derived calli during long-term cultivation.

H3K27me2, an epigenetic mark decorating silenced genes, is not, in contrast to H3K27me3, in the foreground of interest of epigenetic studies. In our experiments, considerable changes in H3K27me2 were detected among samples, including different loading at H3.1 and H3.3 variants (Figure 2B,C,E,F). An increase in the H3K27me2 in cS (Figure 2B) was clearly dictated by the change of this modification at H3.1 (Figure 2C). Higher levels of H3.1K27me2 compared to H3.3K27me2 were detected in all samples (Figure 2C; Appendix A). Interestingly, even higher and mutually comparable H3K27me2 levels were detected in 1y-cS and 1y-cR (Figure 2E), again due to the considerable increase in this mark at the H3.1 variant (Figure 2F) and despite significantly increased H3.3 representation (Figure 2D). On the other hand, it is evident that H3.3K27me2 in 7ds and 1y-cR and 1y-cS are comparable, while H3.1K27me2 in 1-year calli increased markedly (Figure 2C,F; Appendix A). When considering approximately comparable levels of H3.1 and H3.3 variants in 1y-cS and 1y-cR (Figure 2D), a significant increase in H3K27me2 in both 1-year calli is apparent. However, the biological significance of this observation is not obvious.

Epigenetically active compounds are frequently used in plant functional studies, including analyses of callus formation and subsequent processes of plant organ regeneration. Regarding the impact of the TSA, contradictory results have been obtained so far. In rice, a supplement of 1 and 10 μM TSA in CIM inhibited callus formation on mature rice embryos [36], and similarly, callus formation was inhibited by 1 μM TSA on *A. thaliana* leaf explants [16]. In contrast, in our experiments, callus initiation was markedly faster in medium supplemented with 0.5 μM TSA and calli were formed uniformly from all parts of the seedlings, whereas on the control medium, callus initiation started on shoots and root tips (Figure 1A). Importantly, we used a lower concentration of the drug (0.5 μM). Contrasting activities of TSA on callus formation dependent on its concentration were previously reported: 0.5 μM TSA significantly induced callus formation on mature wheat embryos, whereas this process was inhibited under exposure to 2.5 μM TSA [37]. In agreement with this and our results, 0.1 μM TSA induced the transformation of *A. thaliana* hypocotyls to callus [38]. Data on the effect of NaB on callus formation are scarce. Recently, the positive effects of NaB, TSA, and azacytidine, an inhibitor of DNA methyltransferases, on lettuce callus formation and propagation were presented [39]. Although in our experiments NaB’s influence on callus formation was less pronounced compared with TSA, NaB clearly sped up the formation of microcalli on the shoot part of the seedlings (Figure 1A).

The presence of HDACi in CIM was also reflected in structural chromatin features analyzed in our study, often in a distinct and sometimes even opposite manner in NaB- and TSA-treated samples. While the impact of NaB on H3.1:H3.3 ratios in 20-day calli derived from both organs was negligible, the impact of TSA was more pronounced, and in TSA-cR, levels of both H3 variants were comparable (Figure 2A). This is in agreement with the general activity of HDACi, which is supposed to lead to a higher level of open chromatin structure.

Similarly, the effect of NaB on histone acetylation and methylation was rather negligible, while TSA significantly affected PTMs status, especially in cR (Figure 4A,B). Changed levels of methylated marks (partially driven by a decrease in the H3.1 abundance) were accompanied by significantly increased acetylation at H4 lysines (Appendix A). This is fully consistent with our previous observation that the impact of TSA on histone acetylation during *A. thaliana* germination was stronger than that of NaB [24]. Distinct effects of TSA and NaB were also observed in the regeneration of wheat explants [37].

Altogether, changes in histone acetylation in calli compared with 7ds were not as significant as expected. In addition, the impact of HDACi, despite their distinct effects on the dynamics of callus formation (Figure 1A), was rather moderate. The distinct effects of HDACi may be related to the multifarious activity of families of histone deacetylases acting on different lysine residues settled on histone tails at specific gene bodies.

Understanding the factors that control epigenetic changes during callus formation is a prerequisite for targeted plant regeneration and the development of crop manipulation technologies. Callus growth is reportedly affected by the dual action of HDACs in terms of increased levels of acetylation on a genomic scale and histone deacetylation at genes associated with the repression of organ identity [16]. In the present study, we show that histone variant replacement plays a substantial role in chromatin remodeling during callus induction and propagation. Changes in histone acetylations and methylations represent an additional fine-tuning mechanism that is important for activation/repression of specific genes. HDACi supported and accelerated the process of epigenetic imprint homogenization in the cR and cS. Signs of calli origins (shoot vs. root) seem to disappear entirely during the 20-day cultivation in the presence of HDACi. When compared across PTMs of all analyzed histone peptides, three clusters of samples formed in hierarchical clustering. The first cluster is formed by 7ds, the second by cR and both NaB-exposed calli, and the third by cS and TSA-exposed calli (Figure 4C). This clustering is unexpected and demonstrates the specific effects of HDACi on callus formation and development. Interestingly, long-term passaging also led to homogenization of methylation marks between cR and cS but the signs of root and shoot origin in the form of distinct acetylation patterns outlasted even after 1-year cultivation (Figure 4D). The future study will aim to examine the epigenetic traits of organs of the next generations of plants regenerating in the absence/presence of HDACi, which might be particularly important for plant biotechnology applications.

## 4. Materials and Methods

### 4.1. Cultivation of Plants and Calli

Seeds of *Arabidopsis thaliana* of the Columbia ecotype were purchased from the Nottingham Arabidopsis Stock Centre (Nottingham, UK). Seeds were ethanol sterilized and plated on plates with half-strength Murashige–Skoog medium (Duchefa Biochemicals, Haarlem, The Netherlands) and 0.8% (*w*/*v*) plant agar. After 3 days of stratification in the dark at 4 °C, seeds germinated for 7 days in the phytotron under short-day conditions (8 h light, 100 mmol/m^2^·s, 21 °C; 16 h dark, 19 °C). Callus cultures originated from shoot or root parts of seedlings using CIM (Murashige–Skoog medium supplemented by 2 mg/L NAA (1-naphthalenacetic acid, Duchefa Biochemicals), 0.2 mg/L BAP (6-benzylaminopurine, Duchefa Biochemicals), 30 g/L sucrose and 0.8% (*w*/*v*) plant agar), and CIM supplemented by 0.5 mM NaB (Sigma-Aldrich, St. Louis, MO, USA) or 0.5 μM TSA (Sigma-Aldrich) and were cultivated in a dark room at 24 °C. Callus development was monitored on the 10th and 20th day after its initiation. For phenotype monitoring, three biological replicates (i.e., calli propagated at different times) were cultivated with the same results. For callus size measurement, images of calli cultivated for 10 and 20 days were taken and the area was quantified using ImageJ (1.54f) software (for each variant, n > 10). After 20 days of cultivation, calli were collected for MS analysis, with the calli originating from root parts and shoot parts of the seedlings separated.

Root- and shoot-derived calli cultivated on CIM medium were propagated for 1 year with sub-culturing once per month.

### 4.2. Nuclei Isolation

Seven-day-old *A. thaliana* seedlings and shoot- and root-derived calli were ground in liquid nitrogen and homogenized in extraction buffer (10 mM NaCl, 10 mM 2-(N-morpholino) ethanesulfonate pH 6.0, 5 mM EDTA, 0.25 M sucrose, 0.6% Triton X-100, 0.2 M spermidine, 100 mM PMSF, 45 mM sodium butyrate, and 20 mM β-mercaptoethanol). The homogenate was filtered through nylon mesh and centrifuged at 3000× *g* and 4 °C for 10 min. The pellet was washed twice with the extraction buffer, resuspended in Percoll buffer (2.4 g of 5× concentrated extraction buffer, 18 g of Percoll from Sigma-Aldrich), and centrifuged at 4000× *g* and 4 °C for 15 min. Nuclei floating on the Percoll buffer surface were collected.

### 4.3. Histone Extraction and Derivatization

Histone extraction and derivatization were performed as described previously [23,40,41]. Nuclei in wash buffer (75 mM NaCl, 10 mM EDTA, 50 mM Tris-HCl pH 8.0) were centrifuged at 3000× *g* and 4 °C for 10 min, incubated in nuclei lysis buffer (50 mM Tris-HCl, 100 mM NaCl, 3 mM EDTA, 1% CHAPS) supplemented with protease, deacetylase, and phosphatase inhibitors (0.1 mM PMSF, 45 mM sodium butyrate, and 10 µL/mL protease inhibitor cocktail (P9599, Sigma-Aldrich) and 10 µL/mL phosphatase inhibitor cocktail (P5726, Sigma-Aldrich)) for 1 h on ice, spun at 10,000× *g* for 5 min, washed with 50 mM Tris-HCl, and spun at 10,000× *g* for 5 min. Histones were extracted from released chromatin with 250 µL of 0.2 M H_2_SO_4_ overnight, spun at 16,100× *g*. Protein concentration was measured using Micro BCA^TM^ Protein Assay Kit. Then, 16 µg of histone extract in 0.2 M H_2_SO_4,_ (pH adjusted with NH_4_OH to 8) was subjected to a chemical derivatization. After, 10 µL of propionylation reagent (10 µL propionic anhydride + 30 µL ACN (1:3)) was vortexed, spun for several seconds (prepared fresh for each reaction), and added immediately to the histone extract; the pH was adjusted to 8–9 by NH_4_OH. The samples were spun shortly and incubated at 37 °C for 20 min and 750 rpm. The samples were concentrated in SpeedVac to 5 µL at 35 °C for 50 min, followed by the second round of chemical derivatization. Then, the samples were diluted with 50% ACN to a final volume of 20 µL, the pH was adjusted with NH_4_OH to 8, followed by trypsin digestion on a YM-10 Microcon filter unit (Merck Millipore, Burlington, MA, USA). The samples were diluted with 300 µL of 8 M urea in 0.1 M Tris-HCl (pH 8.5), placed in filter unit, centrifuged 14,000× *g* at 20 °C for 30 min, and washed two times with 200 µL of 8 M urea and 100 µL of 100 mM ammonium bicarbonate (AB). Then, 50 µL of 100 mM AB was added with 400 ng of SOLu-Trypsin Dimethylated (Merck) and incubated at 37 °C overnight. The samples were spun at 14,000× *g* for 10 min three times, the second and third time with the addition of 50 µL of 100 mM AB. The samples were concentrated in SpeedVac to 20 µL at 37 °C for 1 h. Newly released peptide N-termini were subjected to chemical derivatization with the same protocol. Then, the samples were concentrated in SpeedVac to dryness at 35 °C overnight. The samples were desalted prior LC-MS/MS with Hypersep Tip C18, 10–200 µL, 60109-209.

### 4.4. LC-MS/MS, Database Search, and MS-Data Evaluation

Propionylated peptides were measured using LC-MS/MS consisting of an Ultimate 3000 RSLC-nano system coupled to an Orbitrap Lumos Tribrid spectrometer (Thermo Fischer Scientific, Waltham, MA, USA) equipped with a Digital PicoView 550 ion source (New Objective) and Active Background Ion Reduction Device (ESI Source Solutions, Woburn, MA, USA). For measurement of peptide samples originating from 7ds and 20-day-old calli, the LC chromatograph was equipped with an X-Bridge BEH 130 C18 trap column (3.5 μm particles, 100 μm ID, 30 mm; Waters), and an Acclaim PepMap100 C18 analytical column (3 μm particles, 75 μm ID, 500 mm; Thermo Fisher Scientific, Waltham, MA, USA). For measurement of peptide samples originating from 1-year-old calli, μPrecolumn C18 PepMap100 trap column (5 μm particles, 300 μm ID, 5 mm; Waters) and Aurora C18 analytical column (1.6 μm particles, 75 μm ID, 25 mm; Ion Opticks) were used.

Prior to LC separation, tryptic digests were online concentrated on trap column. The mobile phase consisted of 0.1% formic acid in water (A) and 0.1% formic acid in 80% acetonitrile (B), with the following proportions of B: 5% to 25% (0–20 min), 25 to 29% (20–30 min), 29 to 32% (30–40 min), 32 to 38% (40–55 min), 38 to 50% (55–75 min), and 50 to 85% (75–85 min), followed by an isocratic wash of 85% B (85–95 min). Equilibration with 99:1 (mobile phase A:B; flow rate 500 nL/min) of the trapping column and the column was done prior to sample injection to sample loop. The analytical column outlet was directly connected to the ion source. MS data were acquired using a data-dependent strategy selecting up top 10 precursors based on precursor abundance in a survey scan (*m*/*z* 350–2000). The resolution of the survey scan was 60,000 with a target value of 4 × 10^5^, one microscan, and maximum injection time of 54 ms. HCD MS/MS spectra were acquired with a target value of 5 × 10^4^ and resolution of 15,000. The maximum injection time for MS/MS was 22 ms. Dynamic exclusion was enabled for 60 s after one MS/MS spectrum acquisition and early expiration was disabled. The isolation window for MS/MS fragmentation was set to 1.6 *m*/*z*.

The raw mass spectrometric data files were analyzed using Proteome Discoverer software (Thermo Fisher Scientific; version 2.2.0.388) with in-house Mascot search engine (Matrix Science, version 2.6.2) to compare acquired spectra with entries in the UniProtKB *A. thaliana* protein database (version 2020_08; 27,500 protein sequences), cRAP contaminant database (downloaded from http://www.thegpm.org/crap/ (accessed on 22 November 2018)), and in-house AT-histone database (version 2017_02; 71 protein sequences). Mass tolerances for peptides and MS/MS fragments were 7 ppm (10 ppm for cRAP) and 0.03 Da (0.5 Da for cRAP), respectively. Semi-Arg-C for enzyme specificity allowing up to two missed cleavages was set. For searches against cRAP and UniProtKB *A. thaliana* databases, the variable modification settings were oxidation (M), deamidation (N, Q), acetylation (K), and propionylation (K, N-term, S, T, Y), while for histone database searches they were acetylation (K), methylation (K, R), dimethylation (K), trimethylation (K), phosphorylation (S, T), propionylation (K, N-term, S, T, Y), and methylation–propionylation (K). Selected histone peptide identifications were manually verified and quantified from the peak areas derived from the EICs using Skyline (64-bit, v. 23.1.1.268 software), including identification alignment across the raw files based on retention time and *m*/*z*. The KNIME Analytics Platform was used for quantitative statistical analysis [24].

The relative abundance of a particular modified peptide form was calculated from the ratio of each individual peptide form peak area to the total area of respective peptide sequence. The peak areas corresponding to post-translationally modified forms of individual histone peptides were treated as compositions and Aitchison’s methodology based on log-ratios was applied in the statistical evaluation [42]. First, areas were transformed to relative abundances (percentages). For comparison of all individual peptide forms, for each peptide, the log2 ratio of relative abundance of one form to the sum of relative abundances of all other forms (alr-transformation of a 2-part composition) was calculated and the *t*-test was applied to assess the difference in each individual form. The significance of between-sample differences was assessed using *t*-tests, setting the significance threshold at *p* < 0.05. In addition, the differences with 0.05 < *p* < 0.1 were interpreted too; such data show important trends in the balance between the levels of epigenetic marks with the opposite or supporting effects on transcription. Note that in compositional data, the relative abundances of individual parts were not directly comparable due to the constant sum constraint leading to a spurious negative correlation. The data analysis was performed in R version 3.6.3 (https://www.R-project.org, accessed on 17 November 2023) using the composition R package (version 1.40-5; https://CRAN.R-project.org/package=compositions, accessed on 17 November 2023) for alr transformation. To uncover inherent patterns within the data, hierarchical clustering was performed. Percentage data were transformed using the pivot coordinate (pivotCoord) method (pivot coordinates as a special case of isometric logratio coordinates) in order to convert the compositional dataset into a suitable format for subsequent clustering. The “dist” function was used to calculate pairwise distances between rows in the “ilr” matrix and the “average” linkage method was chosen to construct the hierarchical tree (dendrogram) of the data.

## Figures and Tables

**Figure 1 plants-12-04177-f001:**
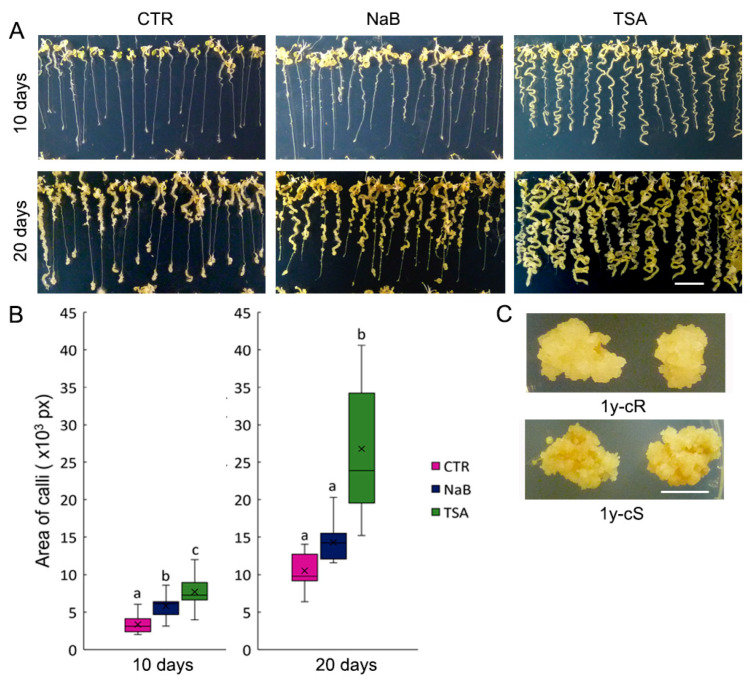
Phenotype of calli originated from Arabidopsis seedlings. (**A**) Formation of callus on 7-day-old *A. thaliana* seedlings on control callus inducing medium CIM (CTR) and CIM supplemented by 0.5 mM sodium butyrate (NaB) or 0.5 μM trichostatin A (TSA). Photos were taken on day 10 (top panels) and day 20 (bottom panels) after the callus initiation. Representative results from three independent experiments are presented. (**B**) Quantification of the areas of calli cultivated for 10 (left) and 20 days (right) is presented by colored box-plots showing means (×) and medians (n > 10). Different letters indicate significant differences based on one-way analysis of variance (ANOVA) with post hoc Tukey test, *p* < 0.01. (**C**) Calli derived from root (1y-cR) and shoot (1y-cS) parts of seedlings propagated for 1 year on the CIM medium. Bar = 1 cm.

**Figure 2 plants-12-04177-f002:**
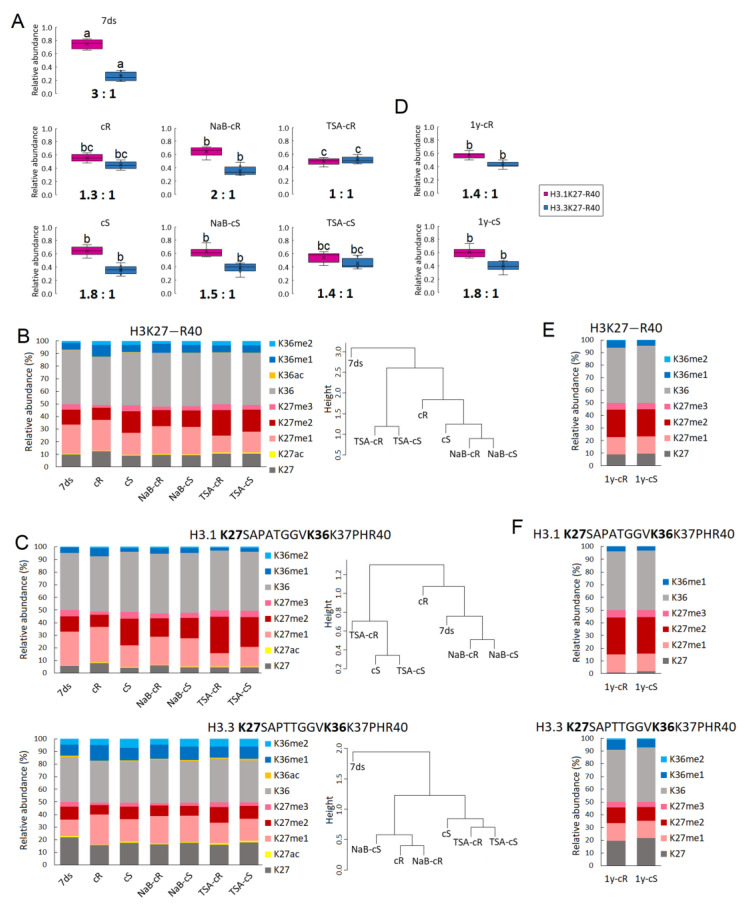
Epigenetic profiles of histone H3 based on H3.1 and H3.3 unique peptides. (**A**) The proportion of H3 variants differed in 7-day-old seedlings (7ds), 20-day calli originated from shoots (cS) and roots (cR), and 20-day calli cultivated in the presence of HDACi (NaB, TSA). Besides the higher level of H3.3 variant in calli compared to seedlings, distinct impacts of NaB and TSA on H3.3 proportion in calli were observed. The H3.1:H3.3 ratios calculated from the median of the replicates in each sample are depicted. Box-plots of H3 proportion obtained from mass spectrometry data show extremes, interquartile ranges, means and medians (n = 5–6). For each histone variant, different letters indicate significant differences between 7ds, control calli, and HDACi-treated calli according to Student’s *t*-test at *p* < 0.05. (**B**) The levels of histone marks at H3K27–R40 peptides differed among 7ds and cS and cR. K27 and K36 marks were quantified regardless of their H3.1 or H3.3 origin while the ratio of H3 variants in each sample was considered. Distinct impacts of NaB and TSA on PTMs in calli were observed. (**C**) The levels of histone marks at H3K27–R40 peptides determined separately for H3.1 and H3.3 variants showed that the differences in PTMs status between cR and cS were mostly driven by distinct K27 and K36 methylations in H3.1 but not in H3.3. The degree of similarity in the PTM profiles of H3.1K27–R40 and H3.3K27–R40 between samples was depicted using hierarchical clustering (distance metric: Euclidean; Clustering method: Average). (**D**) The proportions of H3 variants in 1-year calli derived from roots (1y-cR) and from shoots (1y-cS) remained at the same level as in respective young calli (cS and cR). The H3.1:H3.3 ratios calculated from median of the replicates in each sample were depicted. Box-plots of H3 proportion obtained from mass spectrometry data show extremes, interquartile ranges, means and medians (n = 5–6). There were no significant differences in H3.1:H3.3 ratios between 1y-cR and 1y-cS according to Student’s *t*-test at *p* < 0.05. (**E**) The levels of histone marks at H3K27–R40 peptides were comparable between 1y-cR and 1y-cS. (**F**) The levels of histone marks at H3.1K27–R40 and H3.3K27–R40 peptides were comparable between 1y-cR and 1y-cS.

**Figure 3 plants-12-04177-f003:**
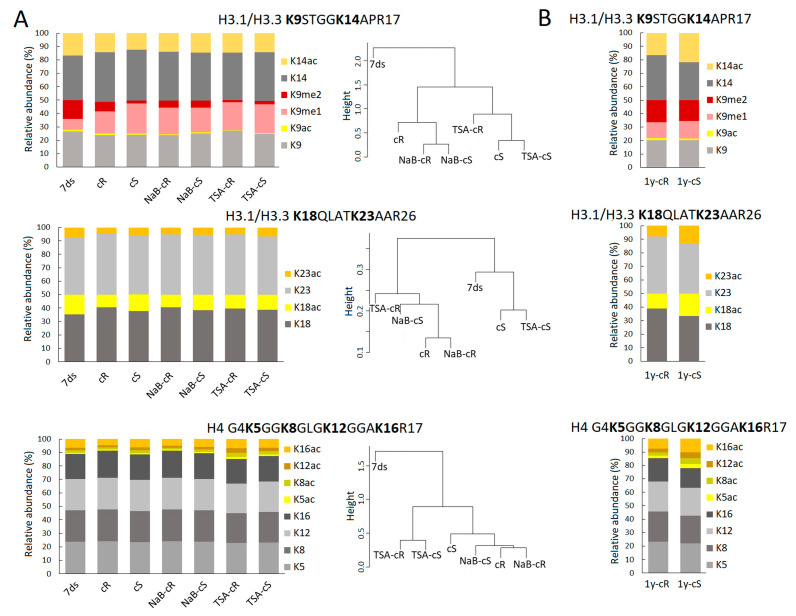
The levels of H3 and H4 histone marks in (**A**) 7-day-old seedlings (7ds), 20-day calli originated from shoots (cS) and roots (cR), 20-day calli cultivated in the presence of HDACi (NaB, TSA), and (**B**) one-year calli originated from shoots (1y-cS) and roots (1y-cS). Distinct impact of NaB and TSA on PTMs of particular histone peptides in calli was observed. The degree of similarity in the PTM profiles of each peptide between samples is depicted using hierarchical clustering (distance metric: Euclidean; Clustering method: Average).

**Figure 4 plants-12-04177-f004:**
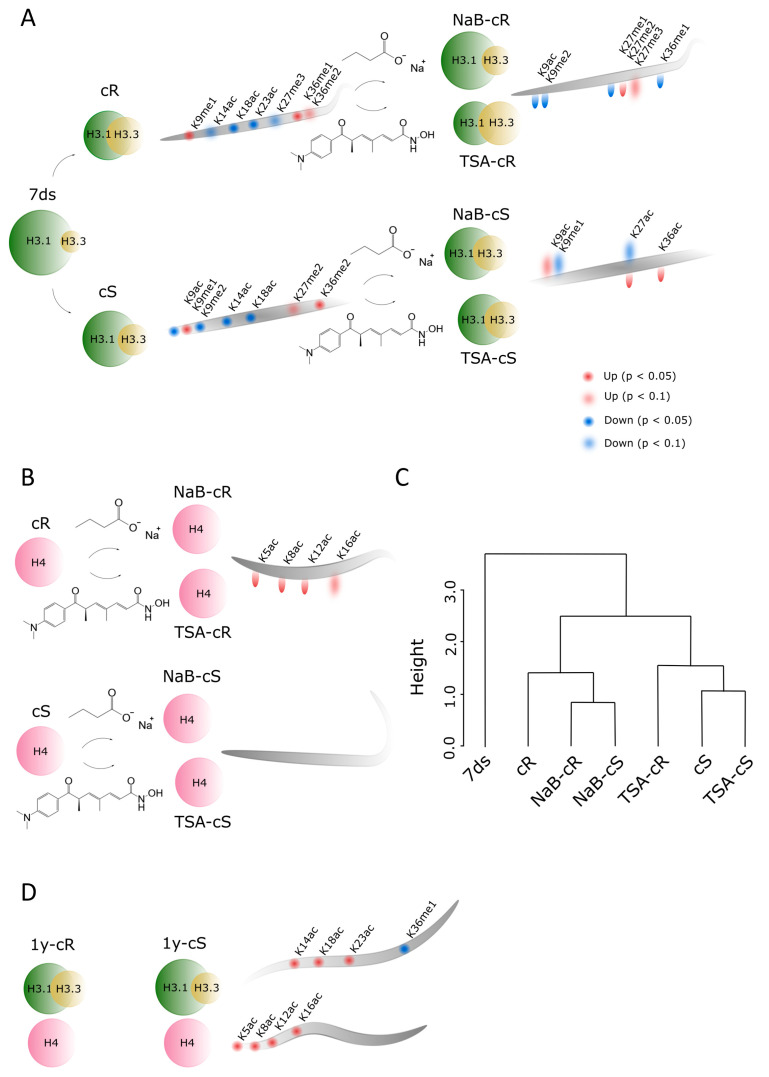
Summary of the most striking features during calli formation. (**A**) Differences in the levels of H3 variants and PTMs between 7ds and shoot- and root-derived calli (cS, cR). Epigenetic changes in calli after HDACi treatment are also depicted, showing higher impact of NaB (indicated by upper comets) on cS than cR, while the impact of TSA (indicated by lower comets) was higher in cR. (**B**) Changes in the levels of PTMs in H4 after HDACi treatment show negligible impact of NaB on H4 acetylations while TSA significantly increased H4 acetylation status. (**C**) Summary dendrogram of hierarchical clustering showing the degree of similarity in the PTM profiles of all analyzed histone peptides among samples (distance metric: Euclidean; Clustering method: Average). (**D**) Differences in the levels of H3 variants and PTMs of H3 and H4 histones between 1-year shoot- and root-derived calli (1y-cS, 1y-cR). Distinct acetylation patterns of H3 and H4 histones in 1y-cS and 1y-cR refer to the different origin of the calli.

## Data Availability

The mass spectrometry proteomics data have been deposited to the ProteomeXchange Consortium via the PRIDE [43] partner repository with the dataset identifier PXD046788.

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
