# Peer review of "Unraveling Epigenetic Changes in A. thaliana Calli: Impact of HDAC Inhibitors"

_plants, 2023, doi:10.3390/plants12244177_

Round 1

Reviewer 1 Report

Comments and Suggestions for Authors

This manuscript describes the histone marks that are present in Arabidopsis calli, and how these marks change in response to extended periods of culture and the application of HDAC inhibitors. Overall I found the manuscript to be interesting with sound methodology. I have some comments that should be addressed prior to publication.

There are only a few corrections that should be made to the English. In the abstract ‘epigenetic changes at the histone proteins’ should be changed to ‘epigenetic changes to histone proteins’. Then the next sentence should read ‘Increased levels of the histone H3.3 variant’.

I would also avoid the use of superlatives such as ‘amazing’ when describing the ability of plants to regenerate (first sentence of introduction).

Scientific/ factual issues: in the first paragraph, the authors state that little or no auxin promotes root differentiation. This is not quite correct: whilst very high levels of auxin promote dedifferentiation and callus formation, lower but still very substantial auxin levels strongly promote root formation coincident with/ immediately following dedifferentiation of tissue. Auxin is also used as a rooting hormone. This should be corrected.

In the third paragraph the authors state that acetylation eliminates negative charges of lysine and arginine residues, but this should be positive charges (their side chains are positively charged).

In paragraph 5 there is a contradiction. It states on the one hand that HDAC mutants did not have altered phenotypes, then in the next sentence they talk about ‘despite severe phenotypic abnormalities….’. This should be clarified.

The concentration of TSA throughout the manuscript seems astonishingly high (molar range). I note there is a large gap between 0.5 and the following M, suggesting a letter is missing, and the concentration should probably be in the micro or nano molar range.

It might be useful to have a close-up of the callus formation on the plants in Fig1, but this is not essential.

In section 2.2, many results are described (such as more or less of a particular H variant PTM) but the p values cited are above 0.05. So it is not always clear which are supposed to be significant and which are not. This needs to be made clearer. I also struggled to find a correlation between the p-values cited in the text and those in the supplementary tables. Can the authors check that the correct values have been used? Overall this comparative analysis section is quite heavy-going, with so many different comparisons made between different samples. It is very good that Fig4 summarises all of these data in a very clear manner.

In section 4.3 the authors describe the PTM inhibitors used but most of these are protease inhibitors and this should be corrected.

Author Response

We would like to thank the reviewer for his/her insightful, helpful and relevant comments on our manuscript.

There are only a few corrections that should be made to the English. In the abstract ‘epigenetic changes at the histone proteins’ should be changed to ‘epigenetic changes to histone proteins’. Then the next sentence should read ‘Increased levels of the histone H3.3 variant’.

Response: Sentences have been adapted as recommended.

I would also avoid the use of superlatives such as ‘amazing’ when describing the ability of plants to regenerate (first sentence of introduction).

 Response: “Amazing capacity of plants to regenerate..” has been replaced by “Exceptional capacity of plants to regenerate…” (l. 31 in the revised ms with highlighted changes).

Scientific/ factual issues: in the first paragraph, the authors state that little or no auxin promotes root differentiation. This is not quite correct: whilst very high levels of auxin promote dedifferentiation and callus formation, lower but still very substantial auxin levels strongly promote root formation coincident with/ immediately following dedifferentiation of tissue. Auxin is also used as a rooting hormone. This should be corrected.

Response: We thank the reviewer for this comment. The sentences have been corrected to correlate with the compositions of CIM, SIM and RIM: “Calli are induced from plant organs/tissues using a callus-inducing medium (CIM) with high auxin and low cytokinin concentrations, shoots are regenerated following the change of the hormone ratio in favor of cytokinins. Roots regenerate on the medium with auxin and low or no cytokinins.”(l. 41-44)

In the third paragraph the authors state that acetylation eliminates negative charges of lysine and arginine residues, but this should be positive charges (their side chains are positively charged).

Response: This is clear mistake – we are sorry about it and many thanks for noticing us.

In paragraph 5 there is a contradiction. It states on the one hand that HDAC mutants did not have altered phenotypes, then in the next sentence they talk about ‘despite severe phenotypic abnormalities….’. This should be clarified.

Response: The phenotypic abnormalities concerned seedlings exposed to the HDAC inhibitors (TSA, NaB), not HDAC mutants. Sentences were rephrased to make it clear (l. 108-111).

The concentration of TSA throughout the manuscript seems astonishingly high (molar range). I note there is a large gap between 0.5 and the following M, suggesting a letter is missing, and the concentration should probably be in the micro or nano molar range.

Response: In our experiments, we exposed calli to 0.5 microM TSA. Unfortunately, during the ms conversion upon its submission, all Greek letters were modified. This has been corrected in the revised version of the ms.

It might be useful to have a close-up of the callus formation on the plants in Fig1, but this is not essential.

In section 2.2, many results are described (such as more or less of a particular H variant PTM) but the p values cited are above 0.05. So it is not always clear which are supposed to be significant and which are not. This needs to be made clearer.

Response: We added a detailed explanation about the significance of differences to the methods: “The significance of between-sample differences was assessed using t-tests, setting the significance threshold at p < 0.05. In addition, the differences with 0.050 < p < 0.1 were interpreted too; such data show important trends in the balance between the levels of epigenetic marks with the opposite or supporting effects on transcription.”(l. 603-607)

I also struggled to find a correlation between the p-values cited in the text and those in the supplementary tables. Can the authors check that the correct values have been used? Overall this comparative analysis section is quite heavy-going, with so many different comparisons made between different samples. It is very good that Fig4 summarises all of these data in a very clear manner.

Response: We agree with the reviewer that this section is “heavy going”. We have tried to improve this section by organizing it better:

  • We put particular comparative analyses to the subsections.
  • We checked all p-values and can confirm that all values were presented correctly in the text. To facilitate navigation to the p-values in supplementary tables, we slightly modified the files, renamed the sheets, and added a reference to the relevant sheet in the text; e.g., l. 194-196: “Relative abundance of H3K27 and H3K36 marks and respective p-values are presented in Supplementary Tables S1 (sheet “2.2.2. H3K27−R40 marks”).

In section 4.3 the authors describe the PTM inhibitors used but most of these are protease inhibitors and this should be corrected.

Response: The term “PTM inhibitors” has been replaced by “protease, deacetylase and phosphatase inhibitors”(l. 526-527).

Reviewer 2 Report

Comments and Suggestions for Authors

The manuscript submitted by Pirek et al. describes epigenetic changes in histones during callus formation from Arabidopsis shoots and roots and the effects of HDAC inhibitors. The authors concluded that epigenetic statements were remarkably similar for callus from roots and NaB treatment, and for callus from shoots and TSA treatment, respectively. The topics described are interesting and important to a wide range of readers. However, I have found some issues as follows that need to be addressed and corrected.

The results revealed by the authors are interesting. Therefore, the authors should discuss the details and possibilities of why epigenetic status is similar between NaB treatment and root callus and between TSA treatment and shoot callus. In addition, why the chromatin is open in callus should be discussed.

Figure 4 seems to summarize well the results of this study. However, I would suggest adding one year of callus data as well.

Overall, most of the changes in histone modifications in callus were increases in activity marks. However, there are also cases of elevated repressive marks such as K27me2. The reasons for this need to be explained and discussed.

Minor comments

Line 147; Figure 2B should be 1B.

It should be written in a unified way, such as HDACi and HDAC inhibitor.

Comments on the Quality of English Language

No comments.

Author Response

We would like to thank the reviewer for his/her insightful, helpful and relevant comments on our manuscript.

 The authors should discuss the details and possibilities of why epigenetic status is similar between NaB treatment and root callus and between TSA treatment and shoot callus. In addition, why the chromatin is open in callus should be discussed.

Response: To be honest, we do not have enough data to speculate on the causes of similarities of cR and NaB-exposed calli, and cS and TSA-exposed calli. We consider this observation very interesting and intend to work on it further. The last para of Discussion has been modified accordingly (l. 470-474).

Articles (ref. 27 – 31) reporting the shift of the chromatin structure to the more relaxed form during the transition from differentiated plant tissue to the callus have been added to the discussion (l. 332-334).

 Figure 4 seems to summarize well the results of this study. However, I would suggest adding one year of callus data as well.

Response: We followed reviewer’s comment and added Figure 4D showing epigenetic status of calli cultivated for one year.

Overall, most of the changes in histone modifications in callus were increases in activity marks. However, there are also cases of elevated repressive marks such as K27me2. The reasons for this need to be explained and discussed.

Response: We are aware of the dynamics of the chromatin modifications during callus formation and propagation. We agree with the referee that the elevated levels of H3K27me2 repressive mark in cS and 1-year-old calli are rather surprising and not quite expected. We added a para to the Discussion dealing with this topic (l. 398-412).

 Minor comments

Line 147; Figure 2B should be 1B.

Response: Corrected.

 It should be written in a unified way, such as HDACi and HDAC inhibitor.

Response: Unified, as recommended.

Reviewer 3 Report

Comments and Suggestions for Authors

The manuscript “Unraveling Epigenetic Changes in A. Thaliana Calli: Impact of HDAC Inhibitors” utlized a mass spectrometry-based approach to investigate epigenetic changes at the histone proteins during callus formation, provided informative data for the understanding of epigenetic control of plant regeneration. Here are some comments:

1 For the phenotype of calli (represented in Fig. 1), the authors need to quantify calli formation, and statistic analyses are required to determine the significance.

2 The merged MS data of root and shoot samples of 7ds was used for comparison with calli derived from root (cR) or shoot (cS), if the authors compare merged MS data of cR and cS with merged MS data of 7ds, is there any similar findings?

Comments on the Quality of English Language The language is readable

Author Response

We would like to thank the reviewer for his/her insightful, helpful and relevant comments on our manuscript.

1 For the phenotype of calli (represented in Fig. 1), the authors need to quantify calli formation, and statistic analyses are required to determine the significance. 

Response: We quantified the dynamics of calli formation, as recommended. Results are presented in Figure 1B.

2 The merged MS data of root and shoot samples of 7ds was used for comparison with calli derived from root (cR) or shoot (cS), if the authors compare merged MS data of cR and cS with merged MS data of 7ds, is there any similar findings?

Response: We followed reviewer’s suggestion and performed comparison between merged data of cR/cS (CTR) and 7ds. We added the following text in paragraph 2.2.:

“To decipher the reason for the low histone yield from roots, we further compared the data of merged cR and cS (CTR) with 7ds, cS, and cR (Supplementary Table S1, sheet “CTR merged data”). Despite the enough starting material, lower overall abundance of cR peptides was detected, and the differences 7ds vs. CTR were found to be more similar to 7ds vs. cS than 7ds vs. cR. Thus, it seems that low root histone yield is related to the nature of the analyzed tissue rather than to the small amount of the input material.”(l. 166-172).